# Stability Evaluation of pH-Adjusted Goat Milk for Developing Ricotta Cheese with a Mixture of Cow Cheese Whey and Goat Milk

**DOI:** 10.3390/foods9030366

**Published:** 2020-03-21

**Authors:** Chung-Shiuan Wu, Jia-Hsin Guo, Mei-Jen Lin

**Affiliations:** 1Department of Food Science, National Pingtung University of Science and Technology, Neipu 91201, Pingtung, Taiwan; oq8501@yahoo.com.tw; 2Department of Animal Science, National Pingtung University of Science and Technology, Neipu 91201, Pingtung, Taiwan

**Keywords:** goat milk, cow cheese whey, delta backscattering profiles, Turbiscan stability index, ricotta cheese

## Abstract

Excess summer milk and a lack of product diversity are major problems facing Taiwan’s dairy goat industry. Gouda and Mozzarella cheeses made with cow milk are popular products for leisure farms in Taiwan, and they produce a large amount of cheese whey as waste. Our objective is to identify the unstable phenomena of pH-adjusted goat milk through the use of Turbiscan Lab^®^ Expert and to produce ricotta cheeses using cow cheese whey waste and excess goat milk. Delta backscattering (∆BS) profiles and the Turbiscan stability index (TSI) were used to evaluate the stability characteristics of goat milk adjusted to pH 6.7–5.2. The results show coagulation phenomena in skimmed goat milk and sedimentation phenomena in full-fat goat milk, when the pH was adjusted to 5.2. The TSI values of goat milk at pH 5.7 and 5.2 were significantly higher (*p* < 0.05) than that of a control. Therefore, 80/20 cow cheese whey/skimmed goat milk and 80/20 cow cheese whey/full-fat goat milk mixtures were acidified to pH 5.5 and heated at 90 °C for 30 min to produce ricotta cheeses A and B. The hardness value, moisture, protein, and ash contents of ricotta cheese A were significantly higher (*p* < 0.05) than that of ricotta cheese B, but no significant difference was found in terms of sensory evaluation.

## 1. Introduction

Goat milk production is ranked third worldwide after cow milk and buffalo milk and is a particularly important dietary component in developing countries [1,2]. Moreover, goat milk is regarded as an indispensable raw material in the dairy industry, due to its high digestive quality, and nutritional value for babies, children, and adults, as well as its beneficial physiological effects for those suffering from malnutrition and indigestion [3,4,5,6]. Goat milk is more prone to sedimentation and coagulation during high heat treatment than cow milk due to its lower α-s1 casein and higher ionic calcium concentrations [1,2,7]. Ethanol stability testing has been widely used to determine the susceptibility of cow and goat milk to heat coagulation [2]. However, the ethanol stability results do not indicate the sedimentation or coagulation of unstable milk during heat treatment.

In 2017, Taiwan produced 386,362 tons of milk from 60,523 cows [8,9], used for producing fresh milk, flavored milk, yogurt, cheese, and other dairy products. Cheese production at leisure farms produces considerable amounts of cheese whey, which is typically discarded as waste. In contrast, in 2017, 45,365 dairy goats produced 14,200 tons of goat milk. Approximately 70% of Taiwan’s dairy goats belong to the Alpine breed [8,9]. In Taiwan, goat milk production is primarily limited to fresh goat milk and flavored goat milk. Furthermore, consumers in Taiwan prefer drinking warm pasteurized goat milk, reducing the demand for goat milk in summer and resulting in considerable surplus production. Developing productive uses for Taiwan’s excess summer goat milk and cow cheese whey is a topic worth exploring.

Ricotta cheese production is considered to be one of the most suitable ways to reuse cheese whey originating from the cheese-making process [10,11]. In addition, ricotta cheese can also be produced from sheep milk, goat milk, or a mixture of cheese whey and milk. Hough et al. [12] indicated that ricotta cheese is a high-moisture soft whey cheese with an initial pH lower than 6.0; consequently, it is highly susceptible to microbial spoilage and has a limited shelf-life, even when refrigerated. Ovine milk and cream can be added to cheese whey to produce whey cheese, increasing the nutrient contents of protein and the fat content of whey cheese. In addition, adding cream creates a softer texture [13]. Kaminarides et al. [14] indicated that whey cheese made from Halloumi cheese whey/full-fat ovine milk has a high sensory score due to its high fat content. However, whey cheese made from Halloumi cheese whey/skimmed ovine milk mixture produces a high nutritional value for consumers. Relatively few studies have examined whey cheese made with a cow cheese whey/goat milk mixture compared to whey cheese made with whole cheese whey, goat cheese whey/goat milk mixture, sheep cheese whey/sheep milk mixture, or cow cheese whey/cow milk mixture. Our research team has successfully developed ricotta cheese made with 100/0, 90/10, and 80/20 ratios of cow cheese whey/cow milk mixtures [15]. Compared to ricotta cheese made with whole cheese whey, the 80/20 cow cheese whey/cow milk was found to be the best ratio for producing ricotta cheese, due to its significantly higher (*p* < 0.05) protein content and yield, scoring higher in terms of overall acceptance from sensory evaluations. Therefore, the ratio of 80/20 cow cheese whey/goat milk is proposed for the production of new dairy products using Taiwan’s excess summer goat milk supply.

This study discusses the sedimentation or coagulation phenomena of pH-adjusted full-fat/skimmed Alpine goat milk from the delta backscattering profiles (∆BS) using Turbiscan Lab^®^ Expert. The Turbiscan stability index (TSI) values calculated from the ∆BS values were used to select an appropriate goat milk pH range for producing ricotta cheese. The study also compares the gross compositions, pH values, titratable acidity, total bacterial counts, texture profile, and sensory evaluation of ricotta cheeses made from cow cheese whey/skimmed goat mixture and cow cheese whey/full-fat goat milk mixture.

## 2. Materials and Methods

The low stability properties of goat milk and the different unstable phenomena resulted in the difficulty associated with developing goat-milk products. In this study, we analyzed the stability characteristics of full-fat/skimmed goat milk of pH 6.7–5.2 with the ∆BS profiles and TSI values at 60 °C in 1 h. The results of the ∆BS profiles and TSI values presented a condition for producing ricotta cheese from goat milk and cow cheese whey. In addition, we envisaged to use the cheese whey and goat milk mixture to develop a ricotta cheese with an eventual objective of providing a high-nutrition dairy product to consumers and improving the diversity of goat milk products in Taiwan.

### 2.1. Preparation of Goat Milk Samples

Goat milk samples were collected from an Alpine Goat Farm in Pingtung, southwest Taiwan. The gross compositions were analyzed using the MilkoScope (Scope Electric^®^, Expert-2059, Regensburg, Germany). Raw goat milk was skimmed at 45 °C using a cream separator (Janschitz GmbH, Milky cream separator, Althofen, Österreich). Hydrochloric acid (1.0 N) was added to both full-fat goat milk (F) and skimmed goat milk (S) to adjust the pH from 6.7 (pH 6.7 F and pH 6.7 S) to 6.2 (pH 6.2 F and pH 6.2 S), 5.7 (pH 5.7 F and pH 5.7 S), and 5.2 (pH 5.2 F and pH 5.2 S). The pH-adjusted goat milk samples were stored at 4 °C for 24 h to allow for ion rebalancing in the goat milk. The pH value, titrate acidity, ionic calcium concentration, and ethanol stability were analyzed at ambient temperature. The delta backscattering profiles and Turbiscan stability index value were analyzed using Turbiscan Lab^®^ Expert at 60 °C for 1 h.

### 2.2. Cheese Whey Collection

Raw cow milk was obtained from a Holstein farm at Pingtung, Taiwan, pasteurized at 63 °C for 30 min in a stainless-steel vat, and quickly cooled to 32 °C. Rennet (0.01%) was added to the pasteurized cow milk to produce cheese curds. After sitting at 32 °C for 30 min, the cheese curds were cut to approximately 1 cm^3^ and allowed to sit again at 32 °C for 30 min. The cheese whey was collected by filtering the cheese curd with a cheese cloth.

### 2.3. Preparation of Ricotta Cheese

Two ricotta cheeses were produced in this study. Cow cheese whey and skimmed/full-fat goat milk mixtures were adjusted to pH 5.5 with citric acid to produce ricotta cheese A (80/20 cheese whey/skimmed goat milk) and B (80/20 cheese whey/full-fat goat milk). The crude fat, lactose, solids-not-fat, crude protein, total solids of ricotta cheese A and B mixtures were 0.25% and 0.96%, 4.19% and 4.11%, 7.63% and 7.48%, 2.80% and 2.75%, and 7.88% and 8.44%, respectively. Both mixtures were heated at 90 °C for 30 min to produce coagulant milk proteins, quickly cooled to 20 °C, collected with a cheese cloth, and drained at 4 °C for 6 h. A 200-g capacity polypropylene cup and a 112-mm diameter polypropylene cap were used to loading the ricotta cheese. The two-ricotta cheese was stored at 4 °C, and we analyzed the gross compositions on the first storage day. Gross compositions, pH, titratable acidity, total aerobic bacterial counts, texture profile, and the sensory evaluation of ricotta cheeses A and B were analyzed at ambient temperature.

### 2.4. Chemical Properties Analysis

The pH values of goat milk and ricotta cheese were measured using a pH meter (Suntex pH/ION meter SP-2500, Taipei, Taiwan). The titratable acidity (%, w/w) of the goat milk and ricotta cheese was determined with 9.0 g of sample mixed with 9.0 g of distilled water titrated against 0.1 N NaOH (Union Chemical Works Ltd., Hsinchu, Taiwan), while a 1% phenolphthalein (Union Chemical Works Ltd., Hsinchu, Taiwan) solution in 95% ethanol (Jiuh Hsing Instruments Co., Ltd., Kaohsiung, Taiwan) was used as the end point indicator [16]. The ionic calcium concentration was measured by the potential difference values (mV) using a calcium ion selective electrode (PASCO, California, USA) attached to a Suntex pH/ION SP-2500 analyzer (Taipei, Taiwan). The ionic calcium concentration (mM) in the goat milk was calculated based on a standard curve of calcium solutions from 1.0 to 20.0 mM. The calcium standards (aqueous CaCl_2_/KCl solutions in an imidazole buffer at pH 6.7) were prepared as previously described [17]. The ethanol stability was determined by mixing equal volumes (1 mL) of goat milk and various concentrations of ethanol (Merck, Darmstadt, Germany; water/ethanol ranging from 10% to 96%, v/v) at 25 °C in a Petri dish. The maximum ethanol concentration in which the goat milk sample did not flocculate was recorded as the ethanol stability value.

### 2.5. Delta Backscattering Profiles and Turbiscan Stability Index Value

The Turbiscan stability index (TSI) was calculated from the delta backscattering (∆BS) values of the pH-adjusted goat milk sample, scanned every 5 min for 1 h at 60 °C in a cylindrical glass cell on the Turbiscan Lab^®^ Expert (Formulaction, Toulouse, France). The light source is an electroluminescent diode in the near infrared range (λair = 880 nm). The variation in the ΔBS signal was calculated as the difference the between backscattering signal at 0 to 60 min, with the ΔBS plotted on the *y*-axis and the sample height (h, mm) on the *x*-axis. The stability parameters were calculated based on the backscattering data, which included the thickness of the formed sediment and cream as the TSI value. The TSI values are represented as TSI-global (TSIG), TSI-bottom (TSIB), TSI-middle (TSIM), and TSI-top (TSIT) to respectively indicate the stability values of the overall (0–40 mm), bottom part (0–10 mm), middle part (10–30 mm), and top part (30–42 mm) of the goat milk samples in the glass cell. Higher TSI values indicate lower sample stability during Turbiscan scanning [18,19].

### 2.6. Ricotta Cheese Gross Compositions

The yield of ricotta cheese was expressed as the fresh weight of the cheese obtained from each liter of cheese whey and goat milk mixture used for production. The moisture was determined at 105 °C through a 1-h drying process according to the method provided in CNS 3443 [20]. The crude fat of ricotta cheese was measured by extraction with petroleum ether in an ANKOM XT10 Extractor (ANKOM Technology, Wayne County, NY, USA). The crude protein of ricotta cheese was determined with the method modified from the CNS 3449 [21]. Ricotta cheese (3.0 g) was accurately weighed, added to a digestion flask, and digested with 20 mL of 98% concentrated sulfuric acid (Union Chemical Works Ltd., Hsinchu, Taiwan). The flask was then connected to a Speed Digestion K-436 (BÜCHI Labortechnik AG, Flawil, Switzerland) and heated at 370 °C for 2 h. The digestion flask was connected to the Distillation Unit K-355 (BÜCHI Labortechnik AG, Flawil, Switzerland). The solution was immediately distilled until the distillate reached approximately 150 mL and was then titrated against 0.1 N sulfuric acid until the end point (i.e., a change in color from green to light red). The total aerobic bacterial count of the ricotta cheese was determined according to CNS 3452 (2007).

### 2.7. Ricotta Cheese Texture Profile Analysis

The texture profile of the ricotta cheese was analyzed in a 200-g plastic cup using a texture analyzer (EZ Test, Shimadzu, Kyoto, Japan) at 10 °C. A 45-mm back extrusion cone probe and a 5-kg load cell were used at 5 mm/s at a depth of 50 mm from the ricotta cheese surface. Averages of five measurements of hardness, cohesiveness, adhesiveness, gum-characteristic, elasticity, and chewing-characteristics were recorded for each sample in individual cups.

### 2.8. Ricotta Cheese Preference Sensory Evaluation

Forty panelists (aged 40–70 years) were recruited from the administrative unit and senior citizens learning camp in National Pingtung University of Science and Technology. Ten grams of ricotta cheese (after 1 day at 4 °C storage) was served in a 20-mL cup labeled with a random three-digit code. The preference sensory evaluation of the ricotta cheese was performed using a nine-point scale, where 1-point corresponds to dislike extremely, 3-point corresponds to dislike, 5-point corresponds to acceptable, 7-point corresponds to like, and 9-point corresponds to like extremely. The following characteristics of ricotta cheese were evaluated before intake: milk aroma, color, and texture; and after intake: hardness, smoothness, granular sensation, and special flavor; as well as overall acceptance.

### 2.9. Statistical Analysis

Statistical analysis was done using Statistical Analysis Software (Version 9.1, SAS Institute Inc., Cary, NC, USA). All experiments were done in triplicates with data expressed as mean ± SD. Triplicate data were subject to analysis of variance (ANOVA) and a *p* value < 0.05 was considered significant. Duncan’s multiple range tests were used to test the significance between experimental groups.

## 3. Results and Discussion

### 3.1. The Gross Composition of Holstein Cow Milk and Alpine Goat Milk

According to the CNS3055 [22], the standards of fat and solids-not-fat (SNF) contents in both raw cow and goat milk must, respectively, exceed 3.0% and 8.0%. The average fat, specific gravity, lactose, SNF, protein, total solids (TS), moisture, and ash contents in Holstein cow milk in this study are shown in Table 1. Walstra et al. [23] indicated that the normal milk of a healthy cow contains 85.3–88.7% moisture, 7.9–10.0% solids-not-fat, 3.8–5.3% lactose, 2.5–5.5% fat, and 2.3–4.4% protein. The gross composition of full-fat and skimmed goat milk in this study were showed in Table 1. Lactose, protein, moisture, ash contents, and specific gravity in skimmed goat milk were significantly higher (*p* < 0.05) than those in full-fat goat milk. Parkash and Jenness [24] indicated that the density of goat milk is similar to that of cow milk. Raynal-Ljutovac et al. [7] showed that the average contents of fat, protein, lactose, and total solids in goat milk were, respectively, 4.0%, 3.5%, 5.0%, and 13%. Zeng et al. [25] suggested the average content of fat, protein, lactose, and SNF in Alpine goat milk were, respectively, 2.45–2.91%, 2.64–3.27%, 4.07–4.44%, and 7.27–8.30%.

### 3.2. The Effect of pH on the Chemical Properties of Full-Fat and Skimmed Alpine Goat Milk

For the control samples, the pH value of raw Alpine goat milk was approximately 6.7 (pH 6.7 S and pH 6.7 F) in the range of 6.6–6.8. The titratable acidity in both pH 6.7 S and pH 6.7 F was 0.15%. The pH of skimmed and full-fat Alpine goat milk was adjusted to 6.2 (pH 6.2 S and pH 6.2 F), 5.7 (pH 5.7 S and pH 5.7 F), and 5.2 (pH 5.2 S and pH 5.2 F) with the addition of 1.0 N HCl and when stored at 4 °C for 24 h to allow for ion rebalancing. Both the skimmed and full-fat Alpine goat milk were unstable and precipitated when the pH was reduced below 5.2 at ambient temperature. The pH values of all the adjusted samples increased slightly after storage at 4 °C, while the titratable acidity increased significantly (*p* < 0.05). Ionic calcium concentrations of skimmed goat milk significantly increased (*p* < 0.05) by 2.79 mM (pH 6.2 S), 6.36 mM (pH 5.7 S), and 13.03 mM (pH 5.2 S), respectively. In addition, ionic calcium concentration of full-fat goat milk significantly increased (*p* < 0.05) by 2.39 mM (pH 6.2 F), 6.29 mM (pH 5.7 F), and 13.12 mM (pH 5.2 F), respectively. However, ethanol stability decreased significantly (*p* < 0.05) by 15%, 27%, and 34% in skimmed goat milk, and by 18%, 27%, and 36% in full-fat goat milk, respectively (Figure 1 and Figure 2). Therefore, the pH value of goat milk decreased after adding 1.0 N HCl, resulting in increased titratable acidity and ionic calcium concentrations, and the ethanol stability decreased.

An increase in hydrogen ions changes the equilibrium of the phosphate and calcium ions in raw goat milk by increasing the dissociation of calcium phosphate after the pH was decreased. The zeta-potential of protein in raw goat milk is also inversely correlated with pH. The increased acidity decreases the negative charge repletion between proteins, causing them to coagulate at a lower percentage of ethanol. When the zeta-potential decreases, the negative charge repletion between proteins increases, preventing protein coagulation and increasing ethanol stability [26,27,28]. Guo et al. [2] reported that ethanol stability decreases below 20% after adjusting the pH of raw goat milk to 5.70 with 2 M HCl. Lin et al. [17] indicated that the concentration of free calcium ions in raw cow milk decreased from 1.78 mM to 1.07 mM after the pH was adjusted from 6.75 to 7.05 with 1 M NaOH. In contrast, this value was increased to 2.59 mM and ethanol stability was decreased below 75% after the pH was decreased to 6.50 with 1 M HCl. Therefore, the physical and chemical properties of goat milk are affected by changes in the pH.

### 3.3. Delta Backscattering Profiles of Full-Fat and Skimmed Alpine Goat Milk at pH 6.7 and pH 5.2

The gross compositions of goat milk are similar to those of cow milk; however, the heat and ethanol stability of goat milk is much lower than that of cow milk, due to its reduced αs1-casein content and consequently higher ionic calcium content in the serum phase [2,7,29]. Lower pH values of skimmed and full-fat goat milk showed lower ethanol stabilities in this study. The ethanol stability test is a practical means of determining the susceptibility of cow milk and goat milk to heat coagulation [2]. However, the ethanol stability results did not result in milk instability during the heat treatments. In this study, the skimmed and full-fat goat milk with different pH values were scanned every 5 min at 60 °C for 1 h to obtain the ∆BS profiles. Changes in the ∆BS profiles indicate sedimentation, coagulation, clarification, or creaming phenomena related to the unstable properties of goat milk samples during scanning; furthermore, the ∆BS profiles were calculated into the TSI values for statistical analysis.

∆BS increases of 5%, 3%, and 18% are, respectively, found at the bottom (height at 0–10 mm), middle (height at 10–30 mm), and top (height at 30–42 mm) parts of pH 6.7 S at 60 °C, when scanned for 1 h, indicating that pH 6.7 S has slight coagulation at the bottom and middle parts but creaming at the top part. ∆BS increased by 24%, 21%, and 25%, respectively, at the bottom, middle, and top parts of pH 5.2 S, indicating that adjusting the pH value of skimmed goat milk to 5.2 resulted in significant coagulation (Figure 3).

The ∆BS change was more complicated and diverse in full-fat goat milk than in skimmed goat milk due to its higher fat content. ∆BS decreased by 4% at the bottom part of pH 6.7 F but increased by 20% at the same part of pH 5.2 F, indicating a slight clarification of pH 6.7 F but a large amount of sedimentation in pH5.2F at 60 °C, when scanned for 1 h. ∆BS increased by 40% at the top part of pH 6.7 F but decreased by 40% at the same part of pH 5.2 F, indicating a large amount of creaming of pH 6.7 F at the top part but a large amount of clarification of pH 5.2 F at the same part (Figure 4). Therefore, the results of the ∆BS profiling indicate the coagulation of skimmed goat milk and the sedimentation of full-fat goat milk when both pH values were adjusted from 6.7 to 5.2.

### 3.4. The Effect of pH on the Turbiscan Stability Index Values of Full-Fat and Skimmed Alpine Goat Milk

Based on the Turbiscan stability profiles, the overall TSI values were determined for all samples (full-fat and skimmed goat milk, pH 6.7–5.2). The ∆BS values were calculated as TSI values, including the TSIG (global), TSIB (bottom), TSIM (middle), and TSIT (top) values, indicating the respective overall changes at the bottom, middle, and top parts of the sample bottle [19]. An increase in the TSI values suggests that the stability of the sample decreases with clarification, creaming, coagulation, or sedimentation.

The TSIG, TSIB, TSIM, and TSIT values of pH 6.7 S, pH 6.2 S, pH 6.7 F, and pH 6.2 F were not significantly different at 60 °C, when scanned for 1 h using Turbiscan. All the TSI values of pH 5.7 S and pH 5.2 S, respectively, increased by 73.2–132.6% and 214.6–419.7% compared to pH 6.7 S. Moreover, all of the TSI values of pH 5.7 F and pH 5.2 F, respectively, increased by 127.2–499.1% and 200.2–723.0%, compared to pH 6.7 F (Table 2). The results of TSIG, TSIB, TSIM, and TSIT show a significant increase (*p* < 0.05), suggesting that the stability of both full-fat and skimmed goat milk at pH 5.7 and 5.2 goat milk decreased dramatically.

Adjusting the pH of skimmed and full-fat goat milk from 6.7 to 5.7 resulted in respective increases to ionic calcium concentrations by 207.1% and 199.7%, and increases to TSIG by 105.7% and 208.3%. In addition, when adjusting the pH of skimmed and full-fat goat milk to 5.2, the increase in the ionic calcium concentrations and TSIG were doubled, compared to pH 5.7. The ethanol stability of pH adjusted skimmed and full-fat goat milk, respectively, decreased from 54% and 57% at pH 6.7 to 27% and 30% at pH 5.7 and, respectively, decreased to 20% and 21% at pH 5.2 (Table 3).

Therefore, a large amount of colloidal calcium was converted to soluble calcium when the pH value of goat milk was lowered, which increased its ionic calcium concentration and decreased the ethanol stability. Adjusting the pH of both skimmed and full-fat goat milk from 6.7 to 5.7 resulted in unstable properties, which further promoted the interaction of κ-casein in goat milk and β-lactoglobulin in cheese whey, in the ricotta cheese production process. Nevertheless, a significant sedimentation was observed when the pH of full-fat goat milk decreased from 6.7 to 5.2, and this was related to the hard texture of the ricotta cheese and the high syneresis of the cheese curd. Therefore, the pH value of goat milk was adjusted to between 5.7 and 5.2 to produce ricotta cheese in this study.

### 3.5. Gross Composition of Ricotta Cheese Made from Goat Milk and Cow Cheese Whey

The mixtures of Ricotta cheese, A and B, were acidified to approximately pH 5.5 with citric acid before 90 °C heat-coagulation. The yields, moisture, crude protein, crude fat, and the ash of ricotta cheese A and B were showed in Table 4. Kaminarides et al. [14] indicated that the increase in the yield of whey cheese could be attributed to the fat content of the mixture adding full-fat goat milk. In this study, there was a slightly higher, but not significantly different, yield of ricotta cheese B over A. In addition, the moisture, crude protein, and ash contents in ricotta cheese A were significantly higher (*p* < 0.05) than in ricotta cheese B (Table 4).

The major proteins of cheese whey were 50% β-lactoglobulin and 20% α-lactalbumin [30,31]. Adding skimmed goat milk to cheese whey facilitates the increase of κ-casein and β-lactoglobulin interaction. Protein contents in skimmed goat milk were higher than in full-fat goat milk, in this study. Thus, the crude protein of ricotta cheese A was significantly higher (*p* < 0.05) than that of ricotta cheese B. The ricotta cheese A contained a significantly higher moisture than cheese B, related to the coagulation phenomena of pH 5.2 S, as shown in Figure 3, indicating that the skimmed goat milk had a higher water-holding capacity than the full-fat goat milk due to the higher protein and lower fat contents in skimmed goat milk.

However, the sedimentation and clarification phenomena of pH 5.2 F, as shown in Figure 4, indicate that full-fat goat milk had higher syneresis properties at pH 5.2 compared to the skimmed goat milk. The characteristics of pH 5.2 F suggest a higher moisture loss in ricotta cheese B than in A during the process. Salvatore et al. [32] indicated that a statistically significant lower moisture content value was observed from the ricotta cheese with a lower protein content, likely due to the water-binding capacity of denatured whey proteins.

Borba et al. [33] presented a ricotta cheese made with 40% cow whey, 40% goat whey, 10% full-fat cow milk, and 10% full-fat goat milk. The resulting yield of ricotta cheese was 7.9%, containing 9.71–10.90% protein and 74% water. Kaminarides et al. [14] indicated that the ash contents of whey cheese made from whey and ovine milk was significantly lower (*p* < 0.05) than the whey cheese made from whey and skimmed ovine milk, due to the higher calcium content in skimmed milk. There were similar results shown in this study.

### 3.6. pH, Ttitratable Acidity, Total Aerobic Bacterial Counts, and Texture Profile of the Ricotta Cheeses

Ricotta cheese has a high moisture content, an initial pH of 6.0, and a limited shelf-life under refrigeration [12]. The total aerobic bacterial counts of ricotta cheeses A and B were, respectively, 2.66 and 2.14, 3.99 and 3.41, 4.76 and 4.25, and 5.75 and 5.33 log cfu/g at the first, fifth, ninth, and 13th day of 4 °C storage (Figure 5). The total aerobic bacterial of ricotta cheese B was significantly lower (*p* < 0.05) than that of ricotta cheese A during storage. The reason for such a rapid increase in the number of total aerobic bacteria was due to the high nutrition of ricotta cheese and the open production space in this study. Hough et al. [12] indicated that a sample with total bacterial counts lower than 5 log cfu/g was recommended to meet food hygiene standards in Argentina. Therefore, applying 90 °C 30 min following container pasteurization to ricotta cheeses A and B resulted in no viable bacterial growth after 48 and 60 days of refrigeration storage, respectively.

According to Taiwan’s Food and Drug Administration, the sanitation regulation for total aerobic bacterial counts in milk and milk products should be lower than 50,000 cfu/g (4.7 log cfu/g) [34]. The total aerobic bacterial counts of ricotta cheeses A and B exceeded 4.70 log cfu/g, respectively, at the ninth and 13th day at 4 °C storage. Figure 5 shows that the estimated shelf life of ricotta cheeses A and B were, respectively, 8 days and 10 days in refrigeration. During 13 days of storage, the pH value of ricotta cheeses A and B were, respectively, 5.79–5.87 and 5.67–5.80, with titratable acidity values of 0.51–0.54% and 0.50–0.53% (Table 5).

Kaminarides et al. [14] indicated that the total bacterial counts were higher in Myzithra cheese made with skimmed ovine milk than in Myzithra cheese made with ovine milk, which could be attributed to contamination during skimming and the addition of skimmed milk to the whey. Hough et al. [12] presented a commercial ricotta cheese made with 40/60 cheese whey/whole milk, showing that the aerobic mesophiles levels were close to 5 log cfu/g, respectively, at the 27th day at 6 °C and at the 8th day at 17 °C. In addition, the pH changes were irregular in both ricotta cheeses stored at different temperatures.

The hardness of ricotta cheese A was significantly higher (*p* < 0.05) than that of ricotta cheese B; however, the adhesiveness was significantly lower (*p* < 0.05) (Table 6). Ricotta cheese A had a harder and stickier texture due to the stronger and more stable structure from its higher protein contents. On the contrary, ricotta cheese B contained 9.45% fat, resulting in a softer texture [13,14]. No significant differences were found in terms of other texture parameters, including cohesiveness, gum-characteristics, elastic, and chewing-characteristics.

### 3.7. Preference Sensory Evaluation of Ricotta Cheese

The preference sensory results of ricotta cheeses A and B were, respectively, 4.83–5.88 and 4.76–6.19 (Figure 6), indicating no significant differences. In addition, the overall acceptance rates of ricotta cheeses A and B were, respectively, 5.56 and 5.68, indicating that both cheeses were deemed acceptable by the panelists. Kaminarides et al. [14] indicated that the Myzithra whey cheese made with whey and full-fat ovine milk mixture had the highest sensory evaluation score, possibly due to its high fat content. Myzithra whey cheese produced with cheese whey and skimmed ovine milk mixture had a hard, dry, granular texture, and a low sensory evaluation score due to the low-fat content. Pizzillo et al. [35] proposed that fat and protein content are the main impact factors for the softness, greasiness, and granularity of the ricotta cheese sensory evaluation. The ricotta cheese with a high fat content produced a soft and greasy texture, while the ricotta cheese with a high protein content produced a granular texture.

## 4. Conclusions

The coagulation in pH 5.2 S and the sedimentation in pH 5.2 F were observed in ∆BS profiles from Turbiscan scanned at 60 °C for 1 h. In addition, both the TSI values of skimmed goat milk and full-fat goat milk at pH values below 5.7 were significantly higher (*p* < 0.05) than at pH 6.7. The pH values of the cow cheese whey/goat milk mixtures were adjusted to 5.5 for ricotta cheese production based on the results of the ∆BS profiles and TSI values. Turbiscan Lab^®^ Expert is a useful tool for providing the analysis of unstable phenomena of pH-adjusted goat milk. Moreover, the moisture, crude protein, ash contents, and hardness in ricotta cheese A were significantly higher than in ricotta cheese B. The higher moisture contents in ricotta cheese A could be explained by the coagulation of pH 5.2 S, which showed a higher water-holding capacity for ricotta cheese made with cow cheese whey and skimmed goat milk. Therefore, developing ricotta cheese using goat milk and cheese whey can effectively help absorb excess summer goat milk production, providing consumers with a new dairy product and increasing the diversity of goat milk products in Taiwan.

## Figures and Tables

**Figure 1 foods-09-00366-f001:**
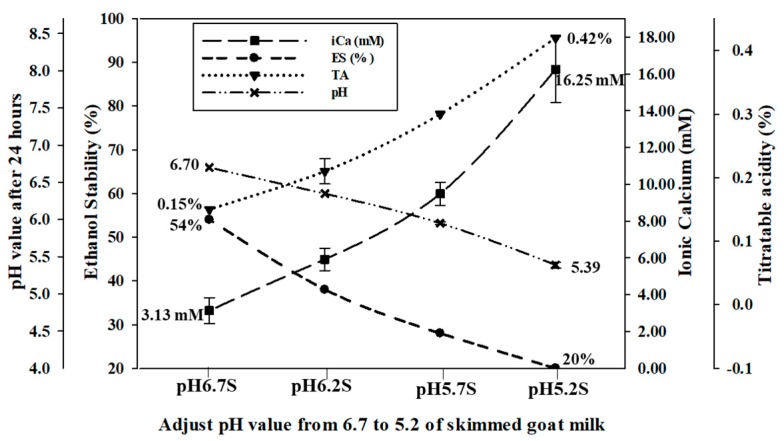
The effect of the pH adjustment of skimmed Alpine goat milk on pH value after 24 h, titratable acidity (TA), ionic calcium concentration (iCa), and ethanol stability (ES).

**Figure 2 foods-09-00366-f002:**
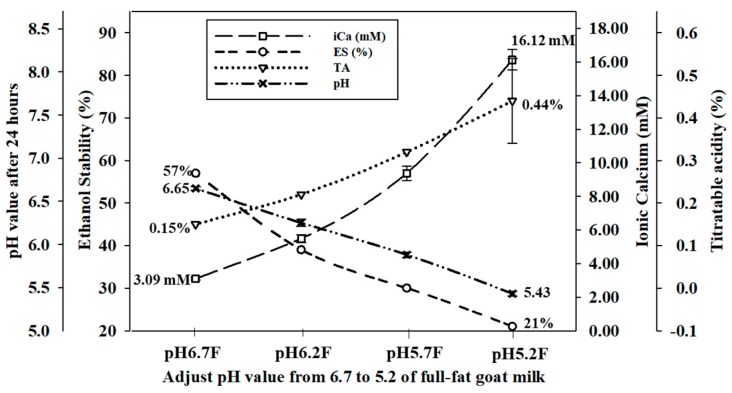
The effect of pH adjustment of full-fat Alpine goat milk on pH value after 24 h, titratable acidity (TA), ionic calcium concentration (iCa), and ethanol stability (ES).

**Figure 3 foods-09-00366-f003:**
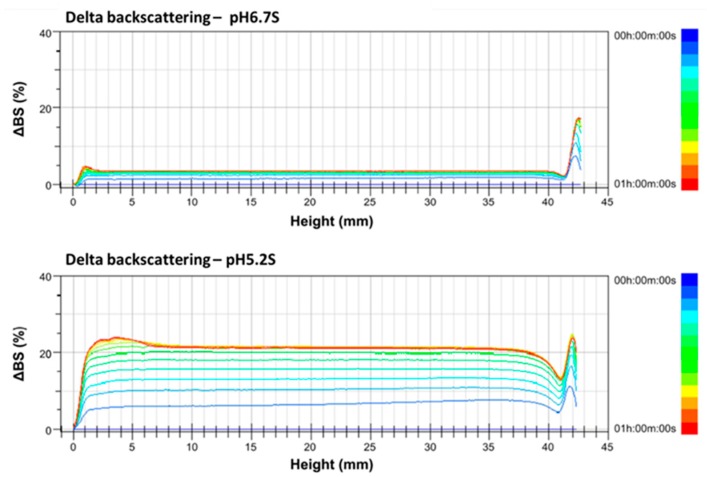
Delta backscattering profiles of pH 6.7 and 5.2 of skimmed Alpine goat milk. The data are reported as a function of time (0–60 min) and sample height (0–42 mm).

**Figure 4 foods-09-00366-f004:**
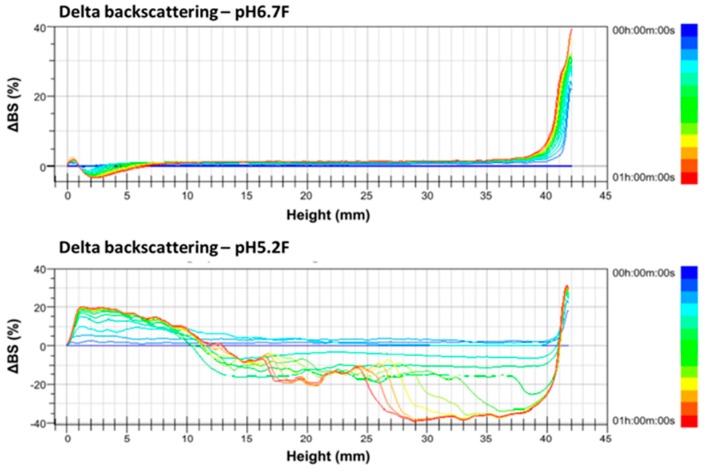
Delta backscattering profiles of pH 6.7 and 5.2 of full-fat Alpine goat milk. The data are reported as a function of time (0–60 min) and sample height (0–42 mm).

**Figure 5 foods-09-00366-f005:**
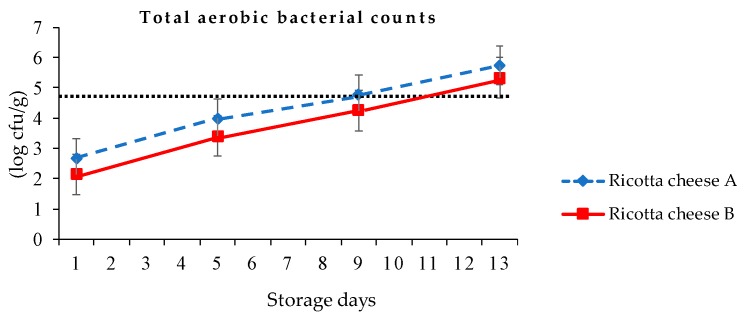
Total aerobic bacterial counts of ricotta cheeses A (80/20 cow cheese whey/skimmed goat milk) and B (80/20 cow cheese whey/full-fat goat milk) at 4 °C storage.

**Figure 6 foods-09-00366-f006:**
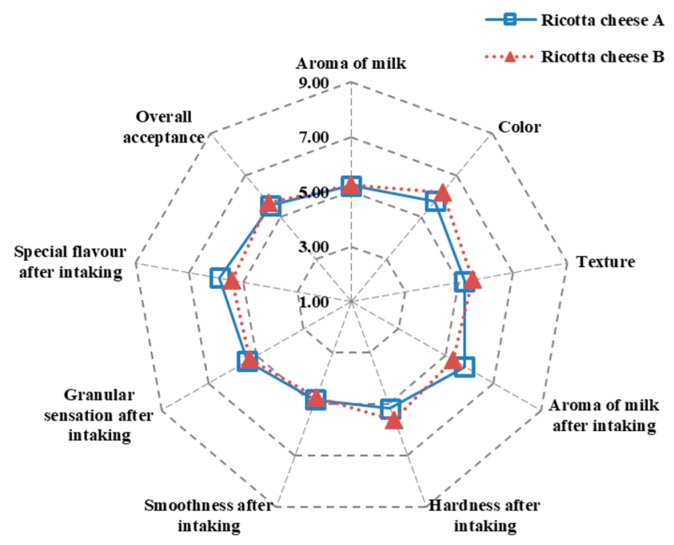
Preference sensory evaluation of ricotta cheeses A (80/20 cow cheese whey/skimmed goat milk) and B (80/20 cow cheese whey/full-fat goat milk).

**Table 1 foods-09-00366-t001:** Compositions of Holstein cow milk and Alpine goat milk (replicated for three different samples).

Compositions	Cow Milk	Full-Fat Goat Milk	Skimmed Goat Milk
Fat (%)	4.17 ± 0.06 ^a^	3.84 ± 0.04 ^b^	0.29 ± 0.02 ^c^
Specific gravity	1.034 ± 0.00 ^b^	1.032 ± 0.00 ^c^	1.037 ± 0.00 ^a^
Lactose (%)	4.73 ± 0.01 ^b^	4.70 ± 0.01 ^b^	5.10 ± 0.03 ^a^
Solid-not-fat (%)	8.62 ± 0.01 ^b^	8.56 ± 0.01 ^b^	9.29 ± 0.06 ^a^
Protein (%)	3.17 ± 0.01 ^b^	3.15 ± 0.01 ^b^	3.40 ± 0.02 ^a^
Total solids (%)	12.80 ± 0.06 ^a^	12.40 ± 0.04 ^b^	9.58 ± 0.07 ^c^
Moisture (%)	87.20 ± 0.06 ^c^	87.60 ± 0.04 ^d^	90.42 ± 0.07 ^a^

Mean ± SE in the same row followed by the same superscripts are not significantly different (*p* < 0.05).

**Table 2 foods-09-00366-t002:** The effect of pH on the Turbiscan stability index value in skimmed and full-fat goat milk (replicated for three different samples). Turbiscan stability index (TSI), TSI-global (TSIG), TSI-bottom (TSIB), TSI-middle (TSIM), and TSI-top (TSIT).

Treatment ^2^	Storage at 4 °C overnight			
	TSIG ^1^	TSIB ^1^	TSIM ^1^	TSIT ^1^
pH 6.7 S(control)	4.07 ± 0.07 ^d^	3.40 ± 0.10d ^e^	3.47 ± 0.13 ^d^	5.33 ± 0.18 ^c^
pH 6.2 S	4.70 ± 0.12 ^d^	4.13 ± 0.18d ^e^	4.20 ± 0.17 ^c,d^	5.73 ± 0.03 ^c^
pH 5.7 S	8.37 ± 0.54 ^c^	7.73 ± 0.56 ^c^	8.07 ± 0.65 ^b^	9.23 ± 0.39 ^b^
pH 5.2 S	17.17 ± 2.23 ^a^	17.67 ± 1.80 ^a^	17.10 ± 2.34 ^a^	16.77 ± 2.50 ^a^
pH 6.7 F(control)	2.53 ± 0.37 ^d^	1.57 ± 0.15 ^e^	1.13 ± 0.27 ^d^	4.93 ± 0.73 ^c^
pH 6.2 F	3.57 ± 0.24 ^d^	2.10 ± 0.15 ^e^	2.30 ± 0.15 ^d^	6.27 ± 0.44 ^c^
pH 5.7 F	7.80 ± 0.15 ^c^	5.40 ± 0.21 ^c,d^	6.77 ± 0.15 ^b,c^	11.20 ± 0.25 ^b^
pH 5.2 F	13.17 ± 0.74 ^b^	11.00 ± 1.78 ^b^	9.30 ± 1.15 ^b^	14.80 ± 0.15 ^a^

^1^ Data are presented as the bottom (TSIB), middle (TSIM), and top (TSIT) of the test bottle, and global (TSIG). ^2^ pH 5.2 S–pH 6.7 F: pH 5.2–6.7 indicates the pH levels of goat milk, S indicates skimmed goat milk, F indicates full-fat goat milk. The mean ± SE in the same column followed by the same superscripts are not significantly different (*p* < 0.05).

**Table 3 foods-09-00366-t003:** Effect of pH on the ionic calcium concentrations, ethanol stability, and TSIG values of skimmed and full-fat goat milk.

Treatment		pH 6.7(Control)	pH 5.7	pH 5.2
Skimmed goat milk	^1^ iCa (mM)	3.09	9.49 (+207.1%)	16.25 (+425.9%)
	^2^ ES (%)	54	27	20
	^3^ TSIG	4.07	8.37 (105.7%)	17.17 (+321.8%)
Full-fat goat milk	^1^ iCa (mM)	3.13	9.38 (+199.7%)	16.12 (+415.0%)
	^2^ ES (%)	57	30	21
	^3^ TSIG	2.53	7.80 (208.3%)	13.17 (+420.6%)

^1^ iCa: ionic calcium concentration. ^2^ ES: ethanol stability. ^3^ TSIG: Turbiscan stability index value on global.

**Table 4 foods-09-00366-t004:** Composition of ricotta cheeses made from goat milk and cheese whey (replicated for three different samples).

Treatment	Yield (%)	Moisture (%)	Crude Protein (%)	Crude fat (%)	Ash (%)
Ricotta cheese A ^1^	6.45 ± 0.57 ^a^	78.83 ± 0.52 ^a^	11.80 ± 0.23 ^a^	2.31 ± 0.24 ^b^	0.66 ± 0.01 ^a^
Ricotta cheese B ^2^	6.95 ± 0.57 ^a^	73.42 ± 0.16 ^b^	9.70 ± 0.04 ^b^	9.45 ± 0.70 ^a^	0.57 ± 0.01 ^b^

^1^ Ricotta cheese made with 80/20 cow cheese whey/skimmed goat mixture. ^2^ Ricotta cheese made with 80/20 cow cheese whey/full-fat goat mixture. Mean ± SE in the same column followed by the same superscripts are not significantly different (*p* < 0.05).

**Table 5 foods-09-00366-t005:** The pH value and titratable acidity of ricotta cheese during storage (replicated for three different samples).

Ricotta Cheese	Days	pH	Titratable Acidity (%)
Ricotta cheese A ^1^	1	5.79 ± 0.00 ^b,c^	0.54 ± 0.01 ^a^
	5	5.77 ± 0.02 ^b,c^	0.53 ± 0.00 ^a,b^
	9	5.78 ± 0.02 ^b,c^	0.51 ± 0.01 ^a,b^
	13	5.87 ± 0.02 ^a^	0.51 ± 0.00 ^a,b^
Ricotta cheese B ^2^	1	5.67 ± 0.01 ^e^	0.50 ± 0.02 ^b^
	5	5.74 ± 0.03 ^c,d^	0.50 ± 0.01 ^a,b^
	9	5.72 ± 0.00 ^d,e^	0.51 ± 0.02 ^a,b^
	13	5.80 ± 0.02 ^b^	0.53 ± 0.00 ^a,b^

^1^ Ricotta cheese made with 80/20 cow cheese whey/skimmed goat mixture. ^2^ Ricotta cheese made with 80/20 cow cheese whey/full-fat goat mixture. The mean ± SE in the same column followed by the same superscripts are not significantly different (*p* < 0.05).

**Table 6 foods-09-00366-t006:** Texture profiles of ricotta cheeses made with cheese whey/full-fat or skimmed goat milk (replicated for three different samples).

**Ricotta Cheese**	**Hardness (N)**	**Cohesiveness**	**Adhesiveness (N)**
Ricotta cheese A ^1^	1.81 ± 0.07 ^a^	0.55 ± 0.02 ^a^	−0.79 ± 0.03 ^b^
Ricotta cheese B ^2^	1.29 ± 0.03 ^b^	0.61 ± 0.03 ^a^	−0.72 ± 0.01 ^a^
**Ricotta Cheese**	**Gum-Characteristic (N)**	**Elastic**	**Chewing-Characteristic (N)**
Ricotta cheese A ^1^	0.89 ± 0.05 ^a^	0.50 ± 0.01 ^a^	0.41 ± 0.02 ^a^
Ricotta cheese B ^2^	0.82 ± 0.02 ^a^	0.54 ± 0.07 ^a^	0.48 ± 0.11 ^a^

^1^ Ricotta cheese made with 80/20 cow cheese whey/skimmed goat mixture. ^2^ Ricotta cheese made with 80/20 cow cheese whey/full-fat goat mixture. The mean ± SE in the same column followed by the same superscripts are not significantly different (*p* < 0.05).

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
