# Peer review of "Stability Evaluation of pH-Adjusted Goat Milk for Developing Ricotta Cheese with a Mixture of Cow Cheese Whey and Goat Milk"

_foods, 2020, doi:10.3390/foods9030366_

Round 1
Reviewer 1 Report
Interesting manuscript requires several corrections.
Comments:
- Avoid quoting in the manuscript the results included in the tables (lines: 164-169, 286, 290-291 etc.)
- Lines 172 and 174 - Sheep's milk is not a test material.
- Lines 188-190 - explain the concentration of calcium ions separately for full-fat milk and separately for skimmed milk.
- Line 190- Ethanol stability also explains separately for full-fat milk and separately for skimmed milk.
- Line 192 – „Therefore, the pH value of goat milk decreased resulting in increased titratable acidity and ionic calcium concentrations, and ethanol stability decreased” - add that the pH value decreased after adding 1.0 N HCl
- Lines 310-316 - Sheep's milk is not a test material. There is a bibliography describing fresh goat cheeses.
- Lines 333. What was the reason for such a rapid increase in the total number of cells of aerobic bacteria? How can this be reduced?
Complete in section 2.6 - storage conditions for cheese: time, temperature, type of packaging, packaging size and tightness, oxygen permeability.
- Line 368. The replay is already on line 150-152. Delete.
Fig.6.- The authors provide a sensory evaluation scale from 1 to 9 (lines 148-149). Figure 6 is made on a scale of 0 to 8 – corrections
- Line 370. How were sensory evaluation calculated? Which study of sensory properties influenced sensory evaluation?
Complete point 2.8 - explain exactly what 1-point for milk aroma means and every 9 points. Describe it for all characteristics: colour, texture, etc.
How was the overall acceptance defined?
Why the value of 5.56 and 5.68 (line 372) is an indicator considered by panellists as acceptable? What value was taken as unacceptable?
Author Response
Dear reviewer
Thank so much for your responding and suggestion. There is the file of revised version manuscript.
- Avoid quoting in the manuscript the results included in the tables (lines: 164-169, 286, 290-291 etc.). Thank you for your suggestion. We have already revised the manuscript as follows: Revised part:(line 171-176, 291-292, 293-296)
- Lines 172 and 174 - Sheep's milk is not a test material. We agree with your response as following: Revised part:(line 177-180)
- Lines 188-190 - explain the concentration of calcium ions separately for full-fat milk and separately for skimmed milk. Thanks for your suggestion. We have already revised the calcium ions results in the manuscript as follows: Revised part:(line 193-196)
- Line 190- Ethanol stability also explains separately for full-fat milk and separately for skimmed milk. Thanks for your suggestion. We have already revised the ethanol stability results in the manuscript as follows: Revised part:(line 196-198)
- Line 192 – „Therefore, the pH value of goat milk decreased resulting in increased titratable acidity and ionic calcium concentrations, and ethanol stability decreased” - add that the pH value decreased after adding 1.0 N HCl. We have added in the manuscript as follows: Revised part:(line 198-200)
- Lines 310-316 - Sheep's milk is not a test material. There is a bibliography describing fresh goat cheeses. We agree with your suggestion. Due to we would like to elaborate our data, we may retain it in the manuscript as follows: Revised part:(line 313-316)
- Lines 333. What was the reason for such a rapid increase in the total number of cells of aerobic bacteria? How can this be reduced? Thank you for the question. We already supplement more experiment results in the manuscript as follows: Revised part:(line 327-332)
- Complete in section 2.6 - storage conditions for cheese: time, temperature, type of packaging, packaging size and tightness, oxygen permeability. Thank you for your suggestion. We supplement more information in the manuscript as follows: Revised part:(line 99-101)
- Line 368. The replay is already on line 150-152. Delete. Thank you for your suggestion. We have already deleted that part.
- Line 370. How were sensory evaluation calculated? Which study of sensory properties influenced sensory evaluation? Complete point 2.8 - explain exactly what 1-point for milk aroma means and every 9 points. Describe it for all characteristics: colour, texture, etc. How was the overall acceptance defined? Why the value of 5.56 and ready 5.68 (line 372) is an indicator considered by panellists as acceptable? What value was taken as unacceptable? Thank you for the suggestion. We already reinterpret the experiment method of sensory evaluation. In this study, we utilize preference sensory evaluation to test the product. If the value was lower than 5.0 that was taken as unacceptable by panellists. We have already supplemented more information in the manuscript as follows: Revised part:(line 152-161)
Thank you very much for your suggestions. Your assistance makes this research more perfect.
Kind regards,
Chung-Shiuan, Wu

Reviewer 2 Report
- 42 …. 45,365 dairy goats in Taiwan
- 90 …..cut to approximately 1 cm3
- 92-99 missing information on A and B composition for origin material (for 80/20 cheese - whey/skimmed goat milk and for 80/20 cheese - whey/full-fat goat milk);
- moreover, I have not found analysis methods for (for example) specific gravity or milk composition …
- 155-158 missing GLM model
- the titratable acidity is not expressed in mmol/l
- 207-208, 210-211 missing information (pH)
- in Conclusion should be information on importance of the results and some possibilities to the future, not repeating of Results and Discussion
Author Response
Dear reviewer
Thank so much for your responding and suggestion. There is the file of revised version manuscript.
- 42 …. 45,365 dairy goats in Taiwan.Thank you for your suggestion. We have already deleted that part.
- 90 …..cut to approximately 1 cm3. Thank you for your suggestion. We have already revised the manuscript as follows: Revised part:(line 90)
- 92-99 missing information on A and B composition for origin material (for 80/20 cheese - whey/skimmed goat milk and for 80/20 cheese - whey/full-fat goat milk); moreover, I have not found analysis methods for (for example) specific gravity or milk composition …Thank you for your suggestion. We already supplement more experiment results in the manuscript as follows: Revised part:(line 95-97, 77-78)
- 155-158 missing GLM model. Thank you for the suggestion. We already reinterpret the experiment method of statistical analysis as follows: Revised part:(line 163-167)
- the titratable acidity is not expressed in mmol/l. Thank you for your suggestion. We already supplement more information as follows Revised part:(line 106)
- 207-208, 210-211 missing information (pH). Thank you for your suggestion. We already supplement that in the manuscript as follows: Revised part:(line 214-215, 217-218)
- in Conclusion should be information on importance of the results and some possibilities to the future, not repeating of Results and Discussion. Thank you for your suggestion. We already reinterpret the conclusion part of this manuscript as follows: Revised part:(line 382-394)
Thank you very much for your suggestions. Your assistance makes this research more perfect.
Kind regards,
Mr. Chung-Shiuan, Wu
